# Universal patterns in egocentric communication networks

Gerardo Iñiguez [1,2,3,4] ✉, Sara Heydari [2], János Kertész [1,5] & Jari Saramäki [2] ✉

Tie strengths in social networks are heterogeneous, with strong and weak ties playing different roles at the network and individual levels. Egocentric networks, networks of relationships around an individual, exhibit few strong ties and more weaker ties, as evidenced by electronic communication records. Mobile phone data has also revealed persistent individual differences within this pattern. However, the generality and driving mechanisms of social tie strength heterogeneity remain unclear. Here, we study tie strengths in egocentric networks across multiple datasets of interactions between millions of people during months to years. We find universality in tie strength distributions and their individual-level variation across communication modes, even in channels not reflecting offline social relationships. Via a simple model of egocentric network evolution, we show that the observed universality arises from the competition between cumulative advantage and random choice, two tie reinforcement mechanisms whose balance determines the diversity of tie strengths. Our results provide insight into the driving mechanisms of tie strength heterogeneity in social networks and have implications for the understanding of social network structure and individual behavior.

Social networks are key to the exchange of ideas, norms, and other cultural constructs in human society[1], influencing the way we communicate[2], support each other[3,4], and form enduring communities[5]. Decades of research have focused on regularities in the patterns of relations among individuals[6] as well as the drivers and mechanisms behind their origin[7]. One particularly prominent feature of social networks is the diversity of tie strengths[8], where strong ties are typically embedded within social groups while weak ties are crucial for the cohesiveness of the network as a whole[8–10]. At the micro level, ego networks—the sets of social ties between an individual (the ego) and their family, friends, and acquaintances (the alters)—commonly feature a small core of close relationships. These close relationships are associated with high emotional intensity and they are surrounded by a larger number of weaker ties. The emergence of this characteristic structural pattern has been associated with

constraints on maintaining social relationships, which include limited information processing capacity[11], social cognition[12–14], and time availability[15–17].

Studies of human communication via mobile phones have shown that in line with the above picture, there is a consistent, general pattern in egocentric networks where a small number of close alters receive a disproportionately large share of communication. Data on the frequency of mobile phone calls and text messages also indicate that within this general pattern, there are clear and persistent individual differences[18–22]: some people repeatedly focus most of their attention on a few close relationships, while others tend to distribute communication among their alters more evenly[18]. These differences are stable in time even under high personal network turnover. However, the mechanisms that generate such heterogeneity of tie strengths, its individual-level variation, and the generality of this pattern beyond

[1]Department of Network and Data Science, Central European University, 1100 Vienna, Austria. [2]Department of Computer Science, Aalto University School of Science, 00076 Aalto, Finland. [3]Faculty of Information Technology and Communication Sciences, Tampere University, 33720 Tampere, Finland. [4]Centro de Ciencias de la Complejidad, Universidad Nacional Autonóma de México, 04510 Ciudad de México, Mexico. [5]Complexity Science Hub, 1080 Vienna, Austria. ✉e-mail: iniguezg@ceu.edu; jari.saramaki@aalto.fi

mobile-phone-mediated communication, have not yet been established[14,22–24].

Here, we explore multiple sets of data on recurring social interactions between millions of people to study heterogeneity in ego network tie strengths and its individual variation, and to shed light on the mechanisms behind this heterogeneity. These large-scale datasets contain metadata on different types of time-stamped interactions, from mobile phone calls to social media, spanning a time range from months to years. They are likely to reflect different aspects of social behavior: e.g., mobile-phone calls between friends, work-related emails, and messages on an Internet forum or dating website serve different purposes and may or may not reflect social relationships that also exist offline. Using social networks reconstructed from the interaction records in our data, we measure the distribution of tie strengths in a massive number of egocentric networks, focusing on how this distribution varies between individuals. We compare observations across several datasets representing different channels of communication and use our observations to construct a minimal, analytically tractable model of egocentric network growth that attributes heterogeneity in tie strengths and its individual variation to the balance between competing mechanisms of tie reinforcement.

We find systematic evidence of broad variation in the distributions of tie strengths in ego networks across all communication channels, including those channels that do not necessarily reflect offline social interactions. The majority of ego networks have heterogeneous tie strengths with varying amounts of heterogeneity, while a minority of individuals distribute their contacts in a homogeneous way. With the help of our model of egocentric network evolution, we attribute the amount of heterogeneity to a mechanism of cumulative advantage[25–27], similar to proportional growth[28] and preferential attachment[29–32]. Homogeneity, in turn, is associated with effectively random choice of alters for communication. The balance between these two mechanisms determines the dispersion of tie strengths in an egocentric network. This balance is captured in our model through a single preferentiality parameter that can be fitted to data for each ego. The distributions of fitted values of this parameter are remarkably similar across different datasets, indicating universal patterns of communication in channels that are very different in nature. Similarly to social signatures[18], we also observe that at the level of individuals, the preferentiality parameter is a stable and persistent indicator of the distinctive way people shape their network on the particular channel.

## Results

We analyze data on recurring, time-stamped social interactions between millions of individuals across 16 communication channels, including phone call records, text messages, emails, and posts from social networks and online forums (Fig. 1). Data include, among others, anonymized metadata for 1.3B calls and 613M messages made by 6M people in a European country during 2007[9,21,33–37], 431k emails by 57k students at Kiel University in 4 months[38,39], and 850k wall posts in Facebook made by 45k users in New Orleans during 2006–2009[39,40]. Periods of observation vary widely, from 1 month of text message logs for 3 mobile phone companies[41] to 7 years of private messages and open forum discussions in the Swedish movie recommendation website Filmtipset[39,42,43] (for data details see Supplementary Information [SI] Section S1, Table S1, and Fig. S1). The analyzed data covers a wide range of population sizes and time scales of activity, and they come from a large enough variety of channels to include typical social contexts of human online communication.

### Tie strengths are heterogeneous and driven by cumulative advantage

The total communication activity $a$ (the number of calls, messages, or posts) between an individual, or ego, and each of the ego's acquaintances, or alters, increases with time (Fig. 1a). Due to variability in communication patterns with different alters, aggregated ego networks typically have heterogeneous activities (or, equivalently, tie strengths). This heterogeneity leads to a broad alter activity distribution $p_a$, defined as the probability that a randomly chosen alter has activity $a$ at the end of the observation period. Following[44], we characterize the spread of $p_a$ by the dispersion index $d = (\sigma^2 - t_r)/(\sigma^2 + t_r)$, where $\sigma^2$ is the variance of $p_a$ and $t_r = t - a_0$ its mean relative to the minimum activity in the ego network (Fig. 1b). We find that in our datasets most egos primarily communicate with a few alters, in agreement with previously observed patterns of mobile phone communication[18,45] and online platform use[46]. These egos have networks with heterogeneous tie strengths, in other words, broad activity distributions $p_a$ with large dispersion $d$, where most events are concentrated on the most communicative alters[18,47] (Fig. 1c and SI Fig. S3). Note that in the following, because of their equivalence, we use the term social signature interchangeably for both individual activity distributions and the activity-rank curves of[18]. In addition to egos with heterogeneous tie strengths, all studied communication channels contain a smaller fraction of egos who distribute their communication more homogeneously among alters, leading to smaller values of $d$ and narrower activity distributions. Indeed, the distribution $p_d$ of the dispersion indices over an entire dataset shows both over-dispersed egos ($d \sim 1$) and egos with more Poissonian social signatures ($d \sim 0$; Fig. 1d and SI Fig. S2). Even egos with similar degrees or strength (total numbers of alters or events) can have heterogeneous or homogeneous activity distributions, which are thus not solely driven by differences in the total level of activity between individuals.

In order to find plausible generative mechanisms behind the diversity of social signatures seen in human communication data, we calculate the probability $\pi_a$ that an alter with current activity $a$ communicates once more with the ego, averaged over all events and alters in the observation period (SI Fig. S4). This measure is akin to the attachment kernel of growing networks[48–50], which has been identified in many cases as a linear function of the degree[51,52], and which has been applied in preferential attachment models[28–30,53]. We further restrict $\pi_a$ to the aggregated data of egos with given values of dispersion $d$ (Fig. 1e and SI Fig. S6). When averaged over heterogeneous egos (large $d$), the connection kernel $\pi_a$ increases monotonically with $a$, indicating cumulative advantage as the way most individuals interact with their acquaintances. Homogeneous egos (low $d$), on the other hand, have a flatter and eventually decreasing kernel closer to the average baseline $\pi_a = \langle 1/k \rangle$ where events are allocated among alters uniformly, which can be modeled by random choice. Despite variations in the ratio of heterogeneous to homogeneous activity distributions across channels (signaled by different shapes of the dispersion distribution $p_d$; Fig. 1f and SI Fig. S2), the connection kernel $\pi_a$ has qualitatively the same functional form for all datasets, and it even has a similar slope for a wide range of activity values (Fig. 1g and SI Fig. S4). The observed increasing kernels are also robust to the degree $k$ of the ego network, with low degrees showing slightly higher levels of cumulative advantage (SI Fig. S5).

### Modeling tie strength heterogeneity

To explore the simplest theoretical mechanisms that may give rise to the observed variability across ego networks, we consider minimal cumulative-advantage dynamics similar to Price's model[26,54], where the probability of communication between an ego and an alter depends on their prior communication activity and a tunable parameter $\alpha$ that modulates random alter choice (Fig. 2). We start with an undirected ego network of degree $k$ where all alters have initial communication activity $a_0$. After $\tau$ interactions, the probability $\pi_a$ that an alter with activity $a$ interacts with the ego at

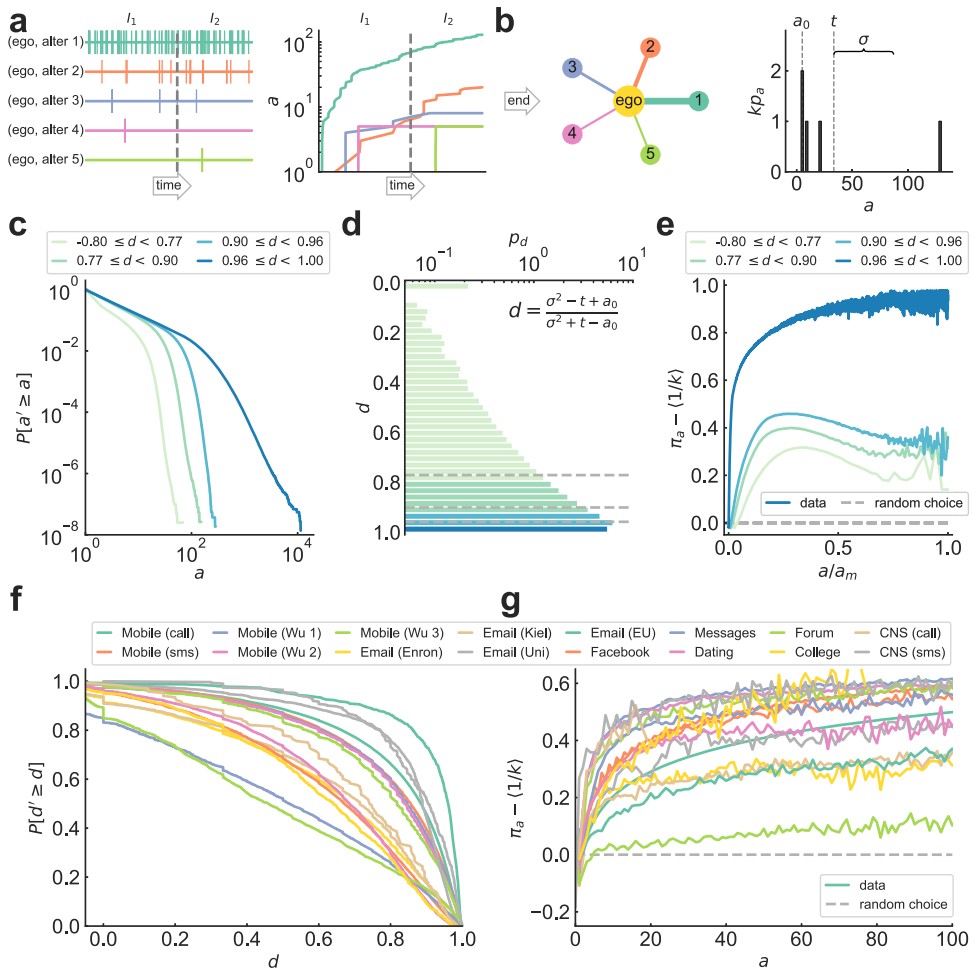

**Fig. 1 | Tie strengths are heterogeneous and driven by cumulative advantage.** **a** Real-time contact sequence between ego and its $k$ alters (left) and timeline of communication activity $a$ (right), for selected ego in the CNS call dataset[75,76] (data description in SI Section S1). Times are relative to the observation length, so close-by events appear as single lines (left) or sudden increases in $a$ (right). The sequence is divided into two consecutive intervals with the same number of events ($l_1$ and $l_2$). With time, some alters communicate more than others. **b** Aggregated ego network (left) and alter activity distribution $p_a$ (right) for (**a**). The distribution has minimum activity $a_0$, mean $t$, and standard deviation $\sigma$. **c** Complementary cumulative distribution function (CCDF) $P[a' \geq a]$ of number of alters with at least activity $a$, for egos in each quartile range of the dispersion distribution $p_d$ and $k \geq 10$, in the Mobile (call) dataset[9,21,33–37] (all systems in SI Fig. S3). For larger dispersions, egos communicate with alters heterogeneously. **d** Dispersion distribution $p_d$ for data in

(**c**), showing more heterogeneous egos (all channels in SI Fig. S2). **e** Relative probability $\pi_a - \langle 1/k \rangle$ that alter with activity $a$ is contacted, averaged over time and egos in each quartile range of the dispersion distribution $p_d$ in (**d**) (all systems in SI Fig. S6). The baseline $\pi_a = \langle 1/k \rangle$ means alters are contacted at random (each $a$ value corresponds to at least 30 egos and is normalized by the maximum activity $a_m$ in the ego subset). For heterogeneous egos, the increasing tendency indicates cumulative advantage: alters with high prior activity receive more events. **f** CCDF $P[d' \geq d]$ of number of egos having at least dispersion $d$, for 8.6M egos in 16 communication channels (SI Table S1 and SI Fig. S2; shown only for egos with more than 10 events). **g** Relative connection kernel $\pi_a - \langle 1/k \rangle$ for all datasets (each $a$ value corresponds to at least 50 egos with $k \geq 2$; see SI Figs. S4–S6). Increasing trends indicate cumulative advantage in all channels.

event time $\tau + 1$ is

$$\pi_a = \frac{a + \alpha}{\tau + k\alpha}. \tag{1}$$

When the parameter $\alpha$ is small, $\pi_a$ increases linearly with activity so egos interact preferentially with the most active alters, following a dynamics similar to stochastic processes driven by cumulative advantage[27,28], and preferential attachment in the evolution of connectivity[29,32,53] and edge weights[30] in growing networks. For large $\alpha$, the connection kernel is flatter and alters are chosen uniformly at random. The parameter $\alpha$ interpolates between heterogeneity and homogeneity in edge weights, even for ego networks with the same mean alter activity $t = \tau/k$ (Fig. 2a; for a detailed model description see Materials and Methods [MM] and SI Section S2).

We solve the model analytically via a master equation for $p_a$ in the limit $\tau, k \to \infty$ (see MM and SI Section S2 for derivation). By introducing

the preferentiality parameter $\beta = t_r/\alpha_r$ with $t_r = t - a_0$ and $\alpha_r = \alpha + a_0$, the activity distribution can be written as

$$p_a = p_0 \frac{a_r^{-1}}{B(a_r, \alpha_r)} \left(1 + \frac{1}{\beta}\right)^{-a_r}, \tag{2}$$

where $a_r = a - a_0$, $p_0 = (1 + \beta)^{-\alpha_r}$, and $B(a_r, \alpha_r)$ is the Euler beta function. Eq. (2) fits to numerical simulations of the model very well, even for relatively low values of $\tau$ and $k$ (Fig. 2b). The preferentiality parameter $\beta$, the ratio between the average number of interactions in the ego network and the tendency of the ego and alters to interact preferentially, reveals a crossover in the behavior of the model, corresponding to a dispersion $d = \beta/(2 + \beta)$ (Fig. 2c; derivation in SI Section S2). For large $\beta$, dispersion increases (just like in the heterogeneous signatures of Fig. 1) and $p_a$ takes the broad shape of a gamma distribution. When $\beta$ and $d$ are small, the activity distribution approaches a Poisson distribution and scales like a Gaussian in the limit

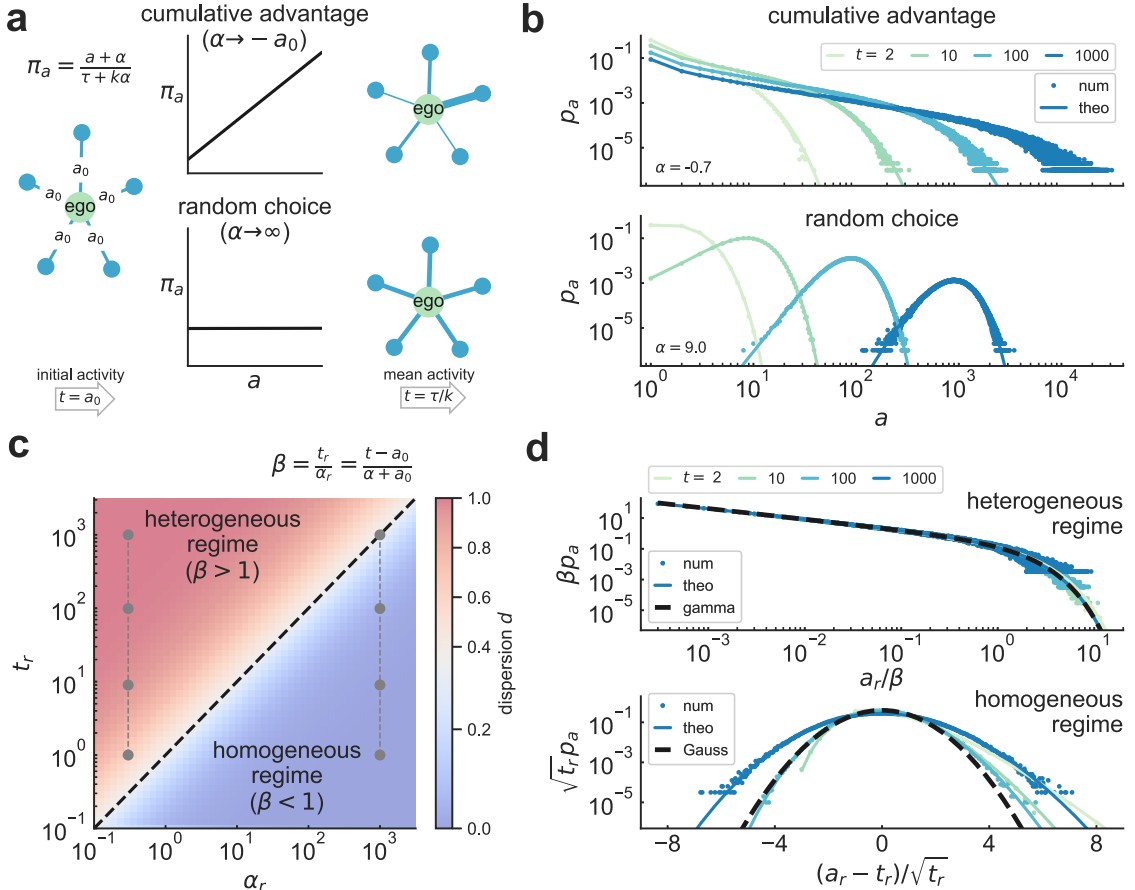

**Fig. 2 | Simple model of alter activity shows crossover in shape of social signatures. a** In a modeled ego network of degree $k$, alters begin with activity $a_0$ and engage in new communication events at event time $\tau$ with probability $\pi_a$, where $a$ is the alter's current activity and $\alpha$ a parameter interpolating behavior between cumulative advantage ($\alpha \to -a_0$, top) and random choice ($\alpha \to \infty$, bottom; see MM and SI Section S2). These dynamics lead to an ego network with mean alter activity (i.e. time) $t = \tau/k$. Plots and networks on the right are shown diagrammatically but correspond to $k = 5$, $a_0 = 1$, $\alpha = -0.9$ ($10^3$), and $t = 3$ ($10^3$) at the top (bottom). **b** Probability $p_a$ that an alter has activity $a$ at time $t$, for varying $t$ with $\alpha = -0.7$ (9) at the top (bottom), $k = 100$ and $a_0 = 1$. Numerical simulations (num) match well with analytical calculations (theo), indicating that cumulative advantage and random choice, respectively, lead to broad or narrow activity distributions. **c** Phase diagram of activity dispersion $d$ in terms of rescaled parameters $\alpha_r = \alpha + a_0$ and $t_r = t - a_0$. The preferentiality parameter $\beta = t_r/\alpha_r$ showcases a crossover between heterogeneous and homogeneous regimes at $\beta = 1$ (dashed line). The vertical gray dash-dotted lines are parameter values for plot (**d**). **d** Rescaled activity distribution $p_a$ for varying $t$ and $\alpha_r = 0.3$ ($10^3$) at the top (bottom). Heterogeneous (homogeneous) regimes show gamma (Gaussian) scaling in $p_a$. All simulations are averages over $10^4$ realizations.

of large $t_r$ (Fig. 2d). This equivalence between $\beta$ and $d$ justifies our choice of the dispersion index as a measure of heterogeneity: $d$ depends only on $\beta$ and allows us to compare egos with different activity levels, while a quantity like the activity variance $\sigma^2 = t_r(1 + \beta)$ depends explicitly on mean activity.

## Model reveals diversity and persistence of social signatures

Empirical ego networks have broadly distributed degree and minimum/mean alter activities for all communication channels studied (see SI Table S1 and Fig. S1). With $k$, $a_0$, and $t$ fixed by the data, Eq. (2) becomes a single-parameter model, allowing us to derive maximum likelihood estimates for the preferentiality parameter $\beta$ in each ego network (Fig. 3; see MM and SI Section S3 for details on the fitting process). After performing a goodness-of-fit test[55–57] with both Kolmogorov-Smirnov and Cramér-von Mises test statistics[58], we obtain $\beta$ estimates for 33–71% of egos in each dataset, amounting to 6.57M individuals over 16 communication channels (SI Tables S2–S3). Values of the preferentiality parameter, capturing the shape of the social signature of an ego, cover a wide region in the ($\alpha_r$, $t_r$) space and accumulate around the crossover $\beta = 1$ (Fig. 3a; compare with Fig. 2c; all datasets in SI Fig. S13). By accumulating all alter activities over heterogeneous ($\beta > 1$) and homogeneous ($\beta < 1$) egos (Fig. 3b and SI

Fig. S14), activity distributions have the same functional form as in Fig. 1c, revealing the crossover value $d = 1/3$ predicted by the model as a principled estimate of the boundary between heterogeneous and homogeneous regimes in Fig. 1c–e.

The heterogeneity of ego network tie strengths is well captured by the preferentiality parameter $\beta$, as it is a single number that encapsulates how each individual chooses which alters to interact with (cumulative advantage or effective random choice). Our data and model show that this parameter is broadly distributed (66–99% of ego networks in a dataset have heterogeneous and 1–34% homogeneous signatures; see SI Table S3). Yet, the parameter has a similar functional shape in data representing different communication channels (Fig. 3c), both in value and in the region in ($\alpha_r$, $t_r$) space covered by data (see SI Fig. S13). To explore whether $\beta$ and the associated activity distribution $p_a$ are personal characteristics of each ego and not a product of random variation, we quantify its persistence by separating the communication activity of an ego into two consecutive intervals[18–21] (with the same number of events; see Fig. 1a), fitting the model independently to each interval. The difference $\Delta\beta$ in preferentiality, relative to $\beta$ for the whole observation period, is very small for most egos (Fig. 3d). When separating individuals by alter turnover in their ego networks, i.e. the Jaccard similarity coefficient $J$ between sets of alters in both intervals,

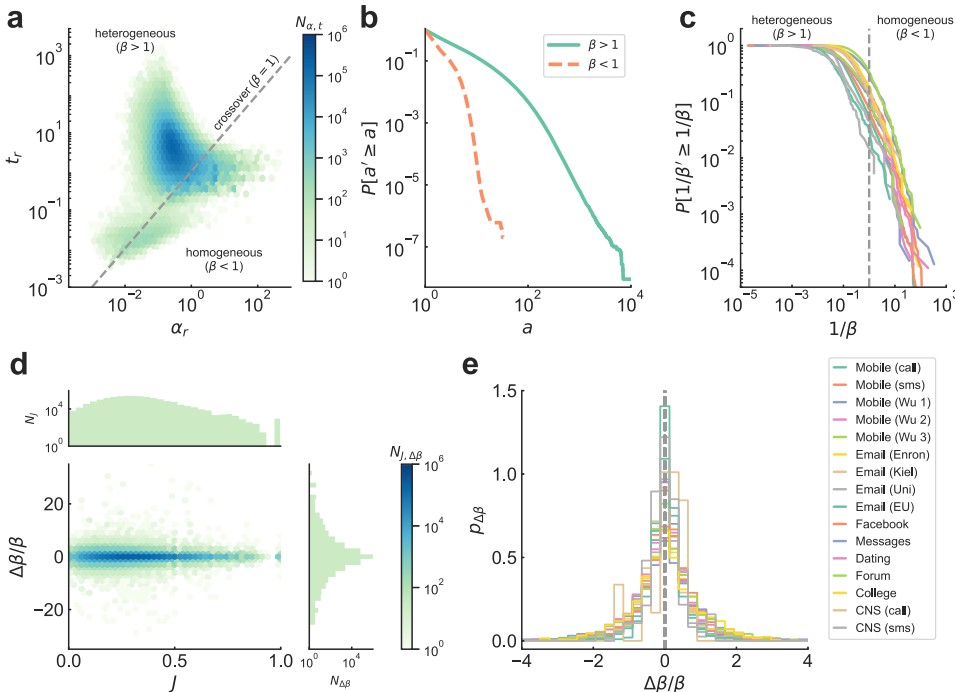

**Fig. 3 | Model reveals diversity and persistence of social signatures. a** Heat map of the number $N_{\alpha,t}$ of egos with given values of $\alpha_r = \alpha + a_0$ and $t_r = t - a_0$ in the Mobile (call) dataset[9,21,33–37] (data description in SI Section S1; all systems in SI Fig. S13). Most egos (95%) have a heterogeneous social signature. On the other side of the crossover $\beta = 1$, a few egos (5%) have more homogeneous tie strengths (SI Table S3). **b** CCDF $P[a' \geq a]$ of number of alters having at least activity $a$, aggregated over all egos in the heterogeneous ($\beta > 1$) or homogeneous ($\beta < 1$) regime in data from (**a**) (all channels in SI Fig. S14). **c** CCDF $P[1/\beta' \geq 1/\beta]$ of rate $1/\beta$, estimated for 6.57M egos in 16 datasets of calls, messaging, and online interactions. All systems show a diversity of social signatures, with 66–99% egos favouring a few of their

alters, and 1–34% communicating homogeneously (SI Table S3 and SI Figs. S11–S12). **d** Number $N_{J,\Delta\beta}$ of egos with given alter turnover $J$ and relative preferentiality change $\Delta\beta/\beta$ when estimating $\beta$ in two consecutive intervals of activity ($I_1$ and $I_2$, see Fig. 1 and SI Section S3), calculated for egos in (**a**) (all channels in SI Fig. S15). We also show marginal number distributions of turnover ($N_J$) and relative preferentiality change ($N_{\Delta\beta}$). Social signatures are persistent in time at the level of individuals, regardless of alter turnover. **e** Distribution $p_{\Delta\beta}$ of relative preferentiality change for all studied datasets. Persistence of social signatures is systematic across communication channels.

the mean of $\Delta\beta$ remains close to zero even for egos with high network turnover ($J \sim 0$; for details see SI Section S3 and SI Fig. S15). The persistence of the preferentiality parameter, found in all of our datasets regardless of communication channel (Fig. 3e) and irrespectively of alter turnover, shows that it indeed captures intrinsic individual differences in social behavior.

## Discussion

Our findings demonstrate that humans tend to build similar-looking personal networks on multiple online communication channels. The analysis of egocentric networks reveals a common heterogeneous pattern, in which a small group of alters receive a disproportionate amount of communication, yet substantial inter-individual variation is observed similarly across all datasets. To capture this pattern and its variation, we have developed a parsimonious and analytically tractable model of ego network evolution, which incorporates a preferentiality parameter specific to each ego. This parameter quantifies the degree of heterogeneity in an ego's personal network, reflecting the balance between two distinct mechanisms of tie reinforcement: cumulative advantage and random choice. Importantly, the distribution of fitted preferentiality parameter values characterizing individual social behavior is consistent across datasets from different channels, pointing to the presence of platform-independent universal patterns of communication.

This universality can be considered both expected and unexpected. In the case of people's real social networks, loosely defined as relationships that exist in the offline world, it is not surprising that their structure, characterized by a small number of close relationships, is reflected in online communication as well, such as through mobile

phone calls. The cumulative advantage mechanism that drives the dispersion of tie strength can be thought to effectively result from people putting more emphasis on their closest relationships, which arise in part due to similarities in any number of sociodemographic, behavioral, and intrapersonal characteristics[59]. Generally, the heterogeneity of tie strengths in ego networks has been attributed to cognitive, temporal, and other constraints[11–13,15–17], and different personality traits[60,61] and their relative stability have been proposed as one possible reason for the persistent individual variation in this heterogeneity[20].

However, there is no a priori reason why the ego networks generated from work-related emails, dating website messages, or movie-related online forum discussions should exhibit similarities to those arising from mobile telephone communications. The nature of communication in these different contexts often pertains to a specific purpose and is limited to a subset of the ego's alters[62], who may even only be represented by online aliases. Nevertheless, despite these differences, the overall pattern of heterogeneous tie strengths and the distribution of the preferentiality parameter, which captures inter-individual variability, are remarkably similar across all datasets. This raises questions as to the underlying mechanisms driving these similarities.

One possibility is that our brain is simply wired to consistently shape our social networks in similar ways, independent of the specific medium of communication[13,63]. Alternatively, the reason may lie in the mechanisms of tie strength reinforcement: cumulative advantage may arise, e.g., because we have already participated in an online conversation with someone and it is easier to continue interacting with the same alter. In other words, while the mechanism of cumulative

advantage effectively explains ego network tie strengths, it can arise because of different reasons: emotional closeness of real relationships, or the ease of repeated interactions in online communication with aliases. A process potentially underlying cumulative advantage is homophily[27,59,64]. If individuals with similar traits communicate more often, as time goes by, alters with large activity will be those most similar to the ego, and also the ones most likely to interact with the ego again, leading to an increasing connection kernel. Random choice and a flat kernel, in turn, are consistent with a lack of similarity-based tie reinforcement. Observational data including individual traits (beyond the activity counts explored here) may allows us to further explore the explicit relationship between cumulative advantage and homophily[65,66].

An alternative perspective to consider is one in which all forms of social connections, whether they occur in-person or virtually, with actual people or pseudonymous entities, are integral components of an egocentric network that encompasses all relationships of an individual. Then, the various communication media can be viewed as distinct dimensions that reflect specific facets of this overarching network. Subnetworks associated with each communication channel are then shaped by the ego's channel preferences and may or may not contain the same alters (see, e.g.,[62]). It is conceivable that the cognitive and time constraints on personal networks act across the whole set of communication channels. Then, each individual has their own way of allocating their available communication activity on the different channels. The selection of a communication channel is known to affect the capacity to sustain emotionally intense social relationships[67], and it is plausible that channel-specific variations in an ego's preferentiality parameter may reflect their ability (or inability) to manage channel-specific constraints that impact effective social bonding. This offers additional insights into the debate surrounding competing theories such as media richness[68] and communication naturalness[63]. Given that the utilized datasets represent distinct populations, it is yet to be determined whether the preferentiality parameter of each individual displays similar or divergent values across different media. Recent research suggests that the values of the preferentiality parameter are similar at least for calls and text messages[21], but it is not certain if this finding generalizes to other channels.

It is also notable that the value of the preferentiality parameter of each ego appears to be stable in time, even in the face of personal network turnover. This suggests that the parameter may reflect a persistent individual trait that influences the structure of egocentric networks on various channels. This interpretation raises important questions about the possible links between an ego's preferentiality parameter and their other personal characteristics, such as age, gender, and health, and whether preferentiality itself is subject to homophilous constraints. It is well established that the diversity of social relationships can serve as an indicator of increased longevity[4], enhanced cognitive functioning during aging[69], and greater resilience to disease[70].

Variation in the preferentiality parameter within a population may have also important consequences at the network level. Egocentric network tie strengths and their variation are obviously related to the well-established heterogeneous distribution of tie strengths across the broader network (see, e.g.,[33]). Moreover, if an ego's parameter value reflects a personal trait, it may also correlate with their network role. For instance, in social media data, personality traits seem to correlate with the ability of an individual to increase their network size[71], broker new relations between alters[72], and participate in more communities[73]. Thus, a broad distribution of preferentiality parameter values among individuals may manifest as a macro-level network structure that reflects a broad array of roles and positions of individuals within the network. These observations highlight the potential for our findings to contribute to a broader understanding of the underlying mechanisms driving social network formation and individual behavior.

## Methods

### Model of alter activity

We consider a minimal ego network dynamics where individuals allocate interactions via cumulative advantage and a tunable amount of random choice (for details see SI Section S2). At initial event time $\tau_0 = k a_0$ with $k$ the degree of the ego network, all alters have minimal activity $a_0$. At any time $\tau \geq \tau_0$, the probability that an alter with activity $a$ becomes active at time $\tau + 1$ is

$$\pi_a = \frac{a_r/t_r + \beta^{-1}}{k(1 + \beta^{-1})}, \tag{3}$$

with $a_r = a - a_0$, $t_r = t - a_0$, and $t = \tau/k$ the mean alter activity. The preferentiality parameter $\beta = t_r/\alpha_r$ (with $\alpha_r = \alpha + a_0$ and $\alpha$ a tunable parameter) interpolates between two regimes: random alter choice ($\beta \to 0$ and $\pi_a \to 1/k$), and preferential alter selection ($\beta \to \infty$ and $\pi_a \to a_r/\tau_r$ with $\tau_r = \tau - \tau_0$).

The model can be treated analytically in the limit $\tau, k \to \infty$ with constant $t$ (SI Section S2). The probability $p_a$ that a randomly chosen alter has activity $a$ follows the master equation

$$d_t p_a = \frac{1}{t + \alpha} \left[ (a - 1 + \alpha) p_{a-1} - (a + \alpha) p_a \right], \tag{4}$$

with initial condition $p_a(a_0) = \delta_{a,a_0}$ and $d_t$ the derivative with respect to $t$. By introducing the probability generating function $g(z,t) = \sum_a p_a z^a$, Eq. (4) reduces to

$$\partial_t g = \frac{z-1}{t+\alpha} (z \partial_z g + \alpha g), \tag{5}$$

a partial differential equation with initial condition $g(z,a_0) = z^{a_0}$. Via the method of characteristics, $g$ takes the explicit form

$$g(z,t) = z^{a_0}[z + (1-z)(1+\beta)]^{-\alpha_r}, \tag{6}$$

from which we obtain the activity distribution $p_a$ in Eq. (2) iteratively by taking partial derivatives of $g$ with respect to $z$. The distribution $p_a$ has mean $t$ and variance $\sigma^2 = t_r(1 + \beta)$, leading to the dispersion index $d = \beta/(2 + \beta)$.

### Fitting data and model

We derive maximum likelihood estimates of the model parameter for empirical ego networks with degree $k$, minimum/maximum alter activity $a_0$ and $a_m$, and total/mean alter activity $\tau = \sum_i a_i$ and $t = \tau/k$ (for details see SI Section S3). Assuming that the $k$ alter activities $\{a_i\}$ are independent and identically distributed random variables following $p_a$ in the model, the likelihood $L_\alpha$ that the sample $\{a_i\}$ is generated by Eq. (2) for given $\alpha$ follows

$$d_\alpha \ln L_\alpha = k \left[ F_\alpha - \ln(1 + \beta) \right], \tag{7}$$

where $F_\alpha = \frac{1}{k} \sum_i [\psi(a_i + \alpha_r) - \psi(\alpha_r)]$ is an average over all observed relative activities $a_r = a_i - a_0$ of the digamma function $\psi(\alpha) = d_\alpha \Gamma(\alpha)/\Gamma(\alpha)$, i.e. the logarithmic derivative of the gamma function $\Gamma(\alpha)$. The $\alpha$ value that maximizes $L_\alpha$ is given implicitly by

$$\alpha_r = \frac{t_r}{e^{F_\alpha} - 1}, \tag{8}$$

or, equivalently, by $\beta = e^{F_\alpha} - 1$.

A goodness-of-fit test allows us to quantify how plausible is the hypothesis that the empirical data is drawn from the model activity distribution in Eq. (2) (SI Section S3). We measure goodness of fit via the standard Kolmogorov-Smirnov statistic

$$D = \max_{a_0 \leq a \leq a_m} |\Delta P_a|, \qquad (9)$$

that is, the largest magnitude of the difference $\Delta P_a(t) = P_{\text{data}}[a' \leq a] - P_a(t)$ between the cumulative distribution of alter activity in data, $P_{\text{data}}[a' \leq a]$, and that of the fitted model, $P_a(t) = \sum_{a' = a_0}^{a} p_{a'}(t)$, across all activities $a \in [a_0, a_m]$. We check the robustness of our results with three other measures from the Cramér-von Mises family of test statistics (for details see SI Section S3).

Given the sample $\{a_i\}$, we compute the estimate $\alpha$ numerically from Eq. (8) and the statistic $D$ from Eq. (9), where the model activity distribution follows Eq. (2). From the model we generate $n_{\text{sim}} = 2500$ simulated activity samples $\{a_i\}_{\text{sim}}$. For each simulated sample, we find its own estimate $\alpha_{\text{sim}}$ and the corresponding statistic $D_{\text{sim}}$. Then, the fraction of simulated statistics $D_{\text{sim}}$ larger than the data statistic $D$ is the $p$-value associated with the goodness-of-fit test, according to $D$. If the $p$-value is large enough ($p > 0.1$ with 0.1 an arbitrary significance threshold), we do not rule out the hypothesis that our activity model emulates the empirical data, and we consider that the ego network has a measurable preferentiality parameter $\beta$. We aim at obtaining large $p$-values (rather than small), since we want to keep the assumption that the model is a good description of the observed data (rather than reject it). Our goodness-of-fit test shows that $33 - 71\%$ of all considered ego networks are well described by the model (or up to $42 - 88\%$ for other test statistics; see SI Table S2).

### Reporting summary
Further information on research design is available in the Nature Portfolio Reporting Summary linked to this article.

## Data availability
For data availability see SI Section S1. Processed data is publicly available at https://github.com/iniguezg/Farsignatures[74]. Raw data is protected and not available due to data privacy laws.

## Code availability
Code to reproduce the results of the paper is publicly available at https://github.com/iniguezg/Farsignatures[74].

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

## Acknowledgements

G.I. thanks Tiina Näsi for valuable suggestions. G.I. and J.K. acknowledge support from AFOSR (Grant No. FA8655-20-1-7020), project EU H2020 Humane AI-net (Grant No. 952026), and CHIST-ERA project SAI (Grant No. FWF I 5205-N). J.K. acknowledges support from European Union's Horizon 2020 research and innovation programme under grant agreement ERC No 810115 - DYNASNET. We acknowledge the computational resources provided by the Aalto Science–IT project. The study was part of the NetResilience consortium funded by the Strategic Research Council at the Academy of Finland (grant numbers 345188 and 345183).

## Author contributions

G.I., S.H., J.K., and J.S. conceived, designed, and developed the study. G.I. and S.H. analyzed empirical data. G.I. derived analytical results and performed numerical simulations and model fitting. G.I., S.H., J.K., and J.S. wrote the paper.

## Competing interests
The authors declare no competing interests.
