## [Peer Review File · Nature Communications]

REVIEWER COMMENTS

Reviewer #2 (Remarks to the Author):

Report on the manuscript "Universal patterns in egocentric communication networks" by Gerardo Iñiguez, Sara Heydari, János Kertész, and Jari Saramäki.

The authors study egocentric communication patterns across an array of large-scale networks of social contacts. Supporting earlier studies on the topic, the authors find heterogeneous activity patterns in ego-centric networks. The core result of this work is the discovery that these heterogeneous communication patterns are universal across all types of communication networks. The authors establish these results through extensive numerical simulations on real data and also using a generative model for the evolution of ego-centric networks. The authors rely on this model to learn the parameters of the egocentric networks. The learned parameters -- particularly, the preferentiality parameter -- exhibit similar statistical patterns in all analyzed networks.

Uncovering patterns and mechanisms of communication in large social networks is one of the central questions in Social and Data Sciences. This work aims to substantially advance both fields through a systematic analysis of a large collection of social networks, focusing on the intensity of social interactions and on network dynamics.

While there is a plethora of experimental works on the topic we lack fundamental insights into the weighted structure of social networks and the dynamics of social networks. By contributing in both directions, the present work aims to fill in the knowledge gap, which justifies its importance for consideration in nature communications.

While I believe that this work makes a good fit for the journal based on its scope and significance, I do have several concerns both about the study and the manuscript.

CONCEPTUAL QUESTIONS:

On the highest level, the storyline of the work goes as follows:

1) the authors study the distributions of activities in ego-centric networks finding heterogeneous interaction activities centered on individual nodes.

2) To explain the observed heterogeneities, the authors make an assumption that social interactions are governed by a mechanism akin to preferential attachment and/or proportional growth. With this assumption in mind, the authors experimentally measure the preferential attachment kernel, finding that the kernel is approximately linear as a function of alter activity.

3) Using the measured linearity, the authors develop a model of preferential dynamics, which they solve analytically for the distribution of node activities.

4) Assuming that social interactions are governed by the model, the authors learn model parameters in ego-centric networks.

5) Authors establish the universality of the learned parameters and conclude the manuscript with a discussion of their findings.

I. The main assumption of this work, is thus, the assumption of the preferential-like kernel governing the ego-centric patterns. Both as a reader and a reviewer, I am looking for a conceptual justification for this assumption. In particular, the preferential attachment kernel assumes the node-centric view, which seems to imply that communication patterns are driven by pure chance. On the other hand, it is also well established that social interactions are driven by similarities: the more similar two nodes are the higher the chance of social interaction. From this standpoint, alters in an ego-centric network are to a certain extent similar to the ego node and one could expect that more similar alter-nodes will attract more activity from the ego node. I wonder how the two principles -- similarity and the preferential attachment -- reconcile in the context of ego-centric networks and invite authors to discuss it in the manuscript.

II. In addition to the conceptual question above, I have some reservations with respect to the authors' conclusion that the attachment mechanism is linear in alter activity. This conclusion seems to follow solely from Fig. 1(e) where authors aggregate all data into a single master curve. Instead of aggregating all data into a single curve for Fig. 1(e), I would suggest splitting the data into several groups based on parameter values and examining the preferential kernels in each of the groups separately. E.g., one could create groups based on the dispersion index (more than two groups, $d < \langle d \rangle$ and $d > \langle d \rangle$), based on k values, and based on the social network analyzed. Can we claim that the linear kernel is observed in most of these groups with different d and k values?

III. I note that different figure panels in the main text correspond to different networks. For instance, Fig. 1a, and b correspond to the CNS network, and Fig. 1c relates to the Forum network. Fig. 3a is related to the Mobile (sms) dataset while Fig. 3b deals with the Facebook network. Ideally, one would showcase a single network in the main text and complement it with other networks in the SI, or showcase all networks everywhere. Otherwise, one might get the impression that authors match their claims with the datasets that better support them, creating a bias in the study.

IV. As a reader, I am curious if all networks exhibit universality not only wrt to the "preferentiality" β parameter but with respect to t_r and a_r parameters. In other words, do all heatmaps in different networks look similar (or different) compared to Fig. 3a? Also, to better assess the universality β parameters, I am curious to see the pdf of its distribution in addition to the ccdf of Fig. 3c. Perhaps, it would fit the Supplementarity Information.

TECHNICAL REMARKS:

While the language and the presentation of the main text seem to be in good order, I nevertheless found the main text somewhat hard to read. I had to switch between the SI and the main text on many occasions to fully understand the results.

1) The formal definition of p_a , the distribution of activities, is only given in the SI. Likewise, probability π_a is not formally defined in the main text.

2) Fig. 1(c) contains two curves per sub-graph labeled as heterogeneous ego and homogeneous ego. After reading the entire manuscript, I am guessing that the two are differentiated based on the $\langle d \rangle$, akin to Fig. 1(d). Is this correct?

3) In Figure 2, the authors introduce heterogeneous and homogeneous regimes based on the β values. These definitions clash with those of heterogeneous ego and homogeneous ego of earlier analysis.

4) If I understand correctly, the main difference between the two plots in Fig. 2b is the value of α . To improve the readability, I invite authors to write corresponding α values in each plot.

In summary, this is a very interesting work with great potential. I would be happy to recommend the manuscript for publication in Nature Communications if my concerns are addressed satisfactorily.

Reviewer #3 (Remarks to the Author):

I have thoroughly enjoyed this manuscript. It is very well written and provides convincing arguments, looking at a classical problem (universal patterns in networks) from a fresh perspective. In my view, the results will be of interest to a broad community interested in networks, as some of the results could prove to be useful in other scientific domains, but also more specifically to researchers in computational social science. The work is complete, and its different aspects, from the data collection and mining to its modelling, are done with care and attention to detail. For these reasons, I am happy to recommend it for publication.

As a minor, optional comment, I would recommend the authors to justify the choice of the dispersion index, taken from ref.46. instead of more usual quantities like the variance.

RESPONSE LETTER FOR

Universal patterns in egocentric communication networks

G. Iñiguez, S. Heydari, J. Kertész, J. Saramäki

We thank the Editor and the two reviewers for a thorough consideration of our work. We have taken all comments into account and implemented them as changes in the attached revised manuscript and SI, highlighted in blue. In what follows we present a point-by-point response to all reviewers' comments, with our responses also highlighted in blue.

As a high-level summary, we have performed the following changes:

Manuscript:

- Updated Fig. 1 (c-e) (dispersion distribution and activity distribution / connection kernel by dispersion group in sample dataset).
- Updated Fig. 3 (a, b, d) (phase diagram, activity distribution by preferentiality, and persistence analysis in sample dataset).
- Updated text in Results and Discussion sections (especially relationship between cumulative advantage and homophily, with updated references).

SI:

- Relationship between activity and dispersion distributions: New Fig. S3 and updated Fig. S2 (and related text in Sec. S1.2).
- Dependence of connection kernel on network degree and activity dispersion: New Figs. S5-S6 (and related text in Sec. S1.2).
- Distribution of preferentiality: New Fig S11 (and related text in Sec. S3.3).
- Activity distribution by preferentiality: New Fig. S14 (and related text in Sec. S3.3).

REVIEWER COMMENTS

Reviewer #2 (Remarks to the Author):

Report on the manuscript "Universal patterns in egocentric communication networks" by Gerardo Iñiguez, Sara Heydari, János Kertész, and Jari Saramäki.

The authors study egocentric communication patterns across an array of large-scale networks of social contacts. Supporting earlier studies on the topic, the authors find heterogeneous activity patterns in ego-centric networks. The core result of this work is the discovery that these heterogeneous communication patterns are universal across all types of communication networks. The authors establish these results through extensive numerical simulations on real data and also using a generative model for the evolution of ego-centric networks. The authors rely on this model to learn the parameters of the egocentric networks. The learned parameters -- particularly, the preferentiality parameter -- exhibit similar statistical patterns in all analyzed networks.

Uncovering patterns and mechanisms of communication in large social networks is one of the central questions in Social and Data Sciences. This work aims to substantially advance both fields through a systematic analysis of a large collection of social networks, focusing on the intensity of social interactions and on network dynamics.

While there is a plethora of experimental works on the topic we lack fundamental insights into the weighted structure of social networks and the dynamics of social networks. By contributing in both directions, the present work aims to fill in the knowledge gap, which justifies its importance for consideration in nature communications.

While I believe that this work makes a good fit for the journal based on its scope and significance, I do have several concerns both about the study and the manuscript.

Response:

We thank the reviewer for a positive assessment of our work, especially the opinion that our work provides fundamental insight into the dynamics of communication in social networks and that it is potentially suitable for publication in Nature Communications.

CONCEPTUAL QUESTIONS:

On the highest level, the storyline of the work goes as follows:

- 1) the authors study the distributions of activities in ego-centric networks finding heterogeneous interaction activities centered on individual nodes.
- 2) To explain the observed heterogeneities, the authors make an assumption that social interactions are governed by a mechanism akin to preferential attachment and/or proportional growth. With this assumption in mind, the authors experimentally measure the preferential attachment kernel, finding that the kernel is approximately linear as a function of alter activity.
- 3) Using the measured linearity, the authors develop a model of preferential dynamics, which they solve analytically for the distribution of node activities.
- 4) Assuming that social interactions are governed by the model, the authors learn model parameters in ego-centric networks.
- 5) Authors establish the universality of the learned parameters and conclude the manuscript with a discussion of their findings.

Response:

We appreciate such a concise description of the storyline of our paper.

I. The main assumption of this work, is thus, the assumption of the preferential-like kernel governing the ego-centric patterns. Both as a reader and a reviewer, I am looking for a conceptual justification for this assumption. In particular, the preferential attachment kernel assumes the node-centric view, which seems to imply that communication patterns are driven by pure chance. On the other hand, it is also well established that social interactions are driven by similarities: the more similar two nodes are the higher the chance of social interaction. From this standpoint, alters in an ego-centric network are to a certain extent similar to the ego node and one could expect that more similar alter-nodes will attract more activity from the ego node. I wonder how the two principles -- similarity and the preferential attachment -- reconcile in the context of ego-centric networks and invite authors to discuss it in the manuscript.

Response:

The reviewer raises the important notion of the relationship between homophily and cumulative advantage, which we now address in the updated Discussion section of our revised manuscript.

As seen in the updated Fig. 1 (c-e) and Figs. S2, S3, S6 in the SI, our first main empirical finding is that ego networks with large dispersion and broad activity distributions have a roughly monotonically increasing connection kernel (implying cumulative advantage), while a minority of ego networks with lower dispersion and narrower activity distributions have flatter and sometimes even decreasing connection kernels (implying random alter choice in communication). While the linear connection kernel of our model (Eq. 1) is indeed an assumption, it is an approximation motivated by this empirical finding across many communication channels.

Now, as the reviewer correctly writes, it is well established that many social interactions (including communication) are driven by similarities, i.e. by homophily. As our data do not contain individual trait information, unfortunately, we cannot directly measure whether alters similar to the ego are more frequently communicated with. However, we have the communication activity counts in time, and in the aggregate, these form an increasing connection kernel for egos with a large dispersion. Therefore, our conceptual justification is that while we cannot directly measure homophily, the observed cumulative advantage mechanism effectively reflects it. If one only sees the communication counts, communication driven by similarity (i.e., homophily) manifests as a cumulative advantage.

A minimal example might clarify our reasoning. Let's take an ego network with two values of an attribute (1, 2) and assume that communication is exclusively driven by homophily, i.e. the ego interacts more often with alters with the same attribute value as the ego (say, 1) and less with the rest (with value 2). Then, as time goes by, alters with attribute 1 will have more activity, and since most alters in a large activity group have attribute 1, the probability that a randomly chosen alter in this group gets contacted again is also higher, implying an increasing connection kernel as a function of activity. So an assumption of homophily-driven communication leads to an aggregate observation of cumulative advantage.

Overall, our interpretation is that cumulative advantage is an effective mechanism that quantifies the preferential accumulation of alter events due to underlying mechanisms, including homophily. Indeed, people choose to communicate due to homophily (and potentially other mechanisms), but in the absence of trait information, we just see the effect of homophily indirectly, or effectively, as a higher probability of interacting with alters that already have more events, i.e. cumulative advantage. Then, the observed

balance between cumulative advantage and random choice is effectively a balance between the presence or absence of such underlying mechanisms. In this sense, communication patterns showing cumulative advantage are less driven by chance, while those with flatter connection kernels are actually more random, at least as far as activity counts are concerned.

We thank the reviewer again for an insightful comment that made us think deeply about the connection between the mechanisms of our model and other principles of social network evolution. We have included a brief account of these arguments in the Discussion, including new references to related work, which we think will open doors for interesting future research.

II. In addition to the conceptual question above, I have some reservations with respect to the authors' conclusion that the attachment mechanism is linear in alter activity. This conclusion seems to follow solely from Fig. 1(e) where authors aggregate all data into a single master curve. Instead of aggregating all data into a single curve for Fig. 1(e), I would suggest splitting the data into several groups based on parameter values and examining the preferential kernels in each of the groups separately. E.g., one could create groups based on the dispersion index (more than two groups, $d < \langle d \rangle$ and $d > \langle d \rangle$), based on k values, and based on the social network analyzed. Can we claim that the linear kernel is observed in most of these groups with different d and k values?

Response:

We agree. We have now separated ego networks into groups corresponding to each of the 4 quartile ranges of both the degree and dispersion distributions and calculated the connection kernel in each. The first observation relates to the kernel's degree dependence (new Fig. S5 in SI). Interestingly, a monotonically increasing connection kernel is robust to the degree of the ego network, appearing in all degree groups and across all datasets. This is consistent with the observation that ego networks with similar degree or strength can have either heterogeneous or homogeneous signatures (end of paragraph 2 in Results), implying that degree is not a defining variable for the regimes of communication behavior we observe. There's a slight tendency of low-degree egos to show more cumulative advantage (the slope of the kernel is larger), which we now mention in the manuscript.

The calculation of the connection kernel by dispersion is quite revealing and consistent with our previous analysis concerning only 2 groups (see new Fig. S6 in SI). High dispersion and broad activity distributions are associated with an increasing kernel and

thus cumulative advantage, while low dispersion and homogeneous activities correspond to flatter and sometimes even decreasing kernels. This is the empirical observation motivating our choice for a cumulative-advantage mechanism in the model. In the model, a linear kernel with progressively larger slope leads to larger dispersion and heterogeneous signatures, just like we observe in communication data.

In light of these results, we have decided to replace Fig. 1 (c-e) with the analysis by dispersion group, since it motivates our modeling choices better. Overall, the increasing nature of the kernel is strongly associated with dispersion, and not so with other network characteristics like degree. We have also modified the text to be more careful in our wording. The kernel in the model is linear (which is an assumption, or an approximation), while the kernel in empirical data of heterogeneous egos is monotonically increasing (beyond fluctuations), with a slope dependent on the activity level. We have kept, in any case, the kernel master curves for all datasets, in both Fig. 1g and Fig. S4 in the SI.

III. I note that different figure panels in the main text correspond to different networks. For instance, Fig. 1a, and b correspond to the CNS network, and Fig. 1c relates to the Forum network. Fig.3a is related to the Mobile (sms) dataset while Fig. 3b deals with the Facebook network. Ideally, one would showcase a single network in the main text and complement it with other networks in the SI, or showcase all networks everywhere. Otherwise, one might get the impression that authors match their claims with the datasets that better support them, creating a bias in the study.

Response:

Agreed. Our original intention was to showcase as many channels as possible in the manuscript while keeping the figures simple, but this approach is better. We have now replaced Fig. 1 (c-e) and Fig. 3 (a, b, d) with results of the Mobile (call) dataset, the largest channel we have. Similar results for the rest of the datasets are correspondingly shown in new/updated Figs. S3, S2, S6, S13, S14, and S15 of the SI. We have kept Fig. 1 (a, b) as is, with a single arbitrarily selected ego from the CNS call dataset, as this is just a diagrammatic example to introduce and give intuition to the relevant measures of our analysis.

To retain some simplicity in the figures, we also note that we have decided to remove the activity-rank distributions (originally in Fig. 1c and Fig. 3b), as this measure is equivalent to the activity distribution p_a and was just included for consistency with previous research in Ref. [18].

IV. As a reader, I am curious if all networks exhibit universality not only wrt to the "preferentiality" β parameter but with respect to t_r and a_r parameters. In other words, do all heatmaps in different networks look similar (or different) compared to Fig. 3a? Also, to better assess the universality β parameters, I am curious to see the pdf of its distribution in addition to the ccdf of Fig. 3c. Perhaps, it would fit the Supplementarity Information.

Response:

We now include the pdf of the beta parameter in the new Fig. S11 of the SI (in addition to the ccdf for all systems in Fig. 3c and Fig. S12), and we refer the reviewer to the (now) Fig. S13 in the SI, which includes the heatmaps of beta values in (α_r, t_r) space for all channels.

Indeed, the channels show universality not only with respect to beta but also with respect to the two parameters defining preferentiality, since these heatmaps look roughly similarly shaped and sized, as a 'u' shape with egos concentrated around the $\beta=1$ crossover line. Larger datasets have a larger area that expands outwards when compared to smaller channels, and the Mobile (call) dataset has a small fraction of egos in the low (α_r, t_r) region as opposed to the rest. But broadly, the similarity across communication channels with such different purposes and behavioral traits is striking, and the main motivation for this study. We have added a line on this to the Results section, when discussing Fig. 3c.

The pdf of the beta parameter in Fig. S11 is also revealing, showing a peak around the crossover $\beta=1$ for most datasets. We mention this finding in related text in Sec. S3.3 of the SI (which connects to the discussion of Fig. 3a in Results).

TECHNICAL REMARKS:

While the language and the presentation of the main text seem to be in good order, I nevertheless found the main text somewhat hard to read. I had to switch between the SI and the main text on many occasions to fully understand the results.

1) The formal definition of p_a , the distribution of activities, is only given in the SI. Likewise, probability π_a is not formally defined in the main text.

Response:

We have rewritten the definitions of p_a and π_a in the Results section of the manuscript, and tweaked the wording of the corresponding definition of π_a in Sec. S1.2 of the SI. We hope the new text clarifies the presentation of our results.

2) Fig. 1(c) contains two curves per sub-graph labeled as heterogeneous ego and homogeneous ego. After reading the entire manuscript, I am guessing that the two are differentiated based on the $\langle d \rangle$, akin to Fig. 1(d). Is this correct?

Response:

As discussed in one of the previous responses, we have now replaced Fig. 1 (c-e) with an analysis of the activity distribution and connection kernel separated by dispersion level, where each dispersion group corresponds to a quartile of the dispersion distribution. From our perspective, this presentation is clearer and removes the need for the labels 'heterogeneous ego' and 'homogeneous ego' in the previous Fig. 1c (where, indeed, they implied the regions $d > \langle d \rangle$ and $d < \langle d \rangle$, respectively).

3) In Figure 2, the authors introduce heterogeneous and homogeneous regimes based on the β values. These definitions clash with those of heterogeneous ego and homogeneous ego of earlier analysis.

Response:

We hope the new presentation of activity distributions and connection kernels by dispersion level (in the new Fig. 1 c, e) makes this point clearer. Empirically (before we make any model assumptions), in Fig. 1 we make the observation that egos with large dispersion and broad activity distributions have an increasing connection kernel, and vice versa for the rest. In this way we can preliminarily (and qualitatively) define heterogeneous egos as those with large $d \sim 1$, and homogeneous egos as those with small $d \sim 0$ (paragraph 3 in Results). Once we introduce the model under the assumption of a linear kernel and analyze it, we formally derive that heterogeneous egos correspond to $\beta > 1$ and an activity distribution scaling like a Gamma distribution, while homogeneous egos have $\beta < 1$ and a p_a scaling like a Poisson or even a Gaussian distribution (for large t_r). The model let us further derive dispersion in terms of β , leading to $d = 1/3$ for $\beta=1$. In other words, assuming the model is valid and cumulative advantage is the mechanism effectively driving communication activity,

$d = 1/3$ is a principled threshold to define heterogeneous and homogeneous egos, which is consistent a posteriori with our initial, qualitative definition in the discussion of Fig. 1. We have rewritten several points in the Results section to make these points clearer, including a line on the connection between beta and d (end of paragraph after discussing Eq. 2).

4) If I understand correctly, the main difference between the two plots in Fig. 2b is the value of α . To improve the readability, I invite authors to write corresponding α values in each plot.

Response:

Agreed. We have updated Fig. 2b as such.

In summary, this is a very interesting work with great potential. I would be happy to recommend the manuscript for publication in Nature Communications if my concerns are addressed satisfactorily.

Response:

We are glad the reviewer finds potential in our paper and is open to a recommendation for publication in Nature Communications. We hope the changes to our manuscript (described above) are enough to address the corresponding concerns.

Reviewer #3 (Remarks to the Author):

I have thoroughly enjoyed this manuscript. It is very well written and provides convincing arguments, looking at a classical problem (universal patterns in networks) from a fresh perspective. In my view, the results will be of interest to a broad community interested in networks, as some of the results could prove to be useful in other scientific domains, but also more specifically to researchers in computational social science. The work is complete, and its different aspects, from the data collection and mining to its modelling, are done with care and attention to detail. For these reasons, I am happy to recommend it for publication.

Response:

We thank the reviewer for a very positive assessment of our work, highlighting its novel perspective and potential contribution to computational social science and other areas. We appreciate the recommendation for publication in Nature Communications.

As a minor, optional comment, I would recommend the authors to justify the choice of the dispersion index, taken from ref.46. instead of more usual quantities like the variance.

Response:

Agreed. We have extended the paragraph after Eq. 2 in the new Results section to clarify this point. Our justification is mostly technical. By defining dispersion d as a ratio involving the variance σ^2 and relative mean t_r of the activity distribution, in the model we can derive it explicitly as $d = \beta / (2 + \beta)$, i.e. dispersion only depends on β and not directly on moments of the activity distribution. This finding highlights that dispersion and preferentiality are in a sense interchangeable and equivalent measures to characterize the regimes of heterogeneity/homogeneity in communication activity. In the model, a more usual quantity like the variance turns out to depend explicitly on the mean, $\sigma^2 = t_r (1 + \beta)$. This reveals another benefit of our definition of dispersion: since d depends only on β , it allows us to compare egos with different activity levels and classify them together as heterogeneous or homogeneous, something we could not do based on activity variance alone.

REVIEWERS' COMMENTS

Reviewer #2 (Remarks to the Author):

My comments were addressed in full in the revised manuscript. Thus, I can now recommend the manuscript for publication.

RESPONSE LETTER FOR

Universal patterns in egocentric communication networks

G. Iñiguez, S. Heydari, J. Kertész, J. Saramäki

REVIEWERS' COMMENTS

Reviewer #2 (Remarks to the Author):

My comments were addressed in full in the revised manuscript. Thus, I can now recommend the manuscript for publication.

We thank the Editor and the reviewers for a thorough consideration of our work, and for the opportunity to publish our work in Nature Communications.